# Vinyl Phosphate-Functionalized, Magnetic, Molecularly-Imprinted Polymeric Microspheres’ Enrichment and Carbon Dots’ Fluorescence-Detection of Organophosphorus Pesticide Residues

**DOI:** 10.3390/polym11111770

**Published:** 2019-10-27

**Authors:** Mao Wu, Yajun Fan, Jiawei Li, Danqing Lu, Yaping Guo, Lianwu Xie, Yiqiang Wu

**Affiliations:** 1College of Sciences, Central South University of Forestry and Technology, Changsha 410004, China; wumaoweiming@126.com (M.W.); fanyajuncsfu@126.com (Y.F.); danqing-1989@163.com (D.L.); guoyaping@csuft.edu.cn (Y.G.); 2College of Material Science and Technology, Central South University of Forestry and Technology, Changsha 410004, China

**Keywords:** pesticide residues, sensor, magnetic molecularly imprinted polymeric microspheres, organophosphorus pesticides, fluorescence detection

## Abstract

The rapid detection of organophosphorus pesticide residues in food is crucial to food safety. One type of novel, magnetic, molecularly-imprinted polymeric microsphere (MMIP) was prepared with vinyl phosphate and 1-octadecene as a collection of dual functional monomers, which were screened by Gaussian09W molecular simulation. MMIPs were used to enrich organic phosphorus, which then detected by fluorescence quenching in vinyl phosphate-modified carbon dots (CDs@VPA) originated from anhydrous citric acid. MMIPs and CDs@VPA were characterized by TEM, particle size analysis, FT-IR, VSM, XPS, adsorption experiments, and fluorescence spectrophotometry in turn. Through the fitting data from experiment and Gaussian quantum chemical calculations, the molecular recognition properties and the mechanism of fluorescence detection between organophosphorus pesticides and CDs@VPA were also investigated. The results indicated that the MMIPs could specifically recognize and enrich triazophos with the saturated adsorption capacity 0.226 mmol g^−1^, the imprinting factor 4.59, and the limit of recognition as low as 0.0006 mmol L^−1^. Under optimal conditions, the CDs@VPA sensor has shown an extensive fluorescence property with a LOD of 0.0015 mmol L^−1^ and the linear range from 0.0035 mmol L^−1^ to 0.20 mmol L^−1^ (R^2^ = 0.9988) at 390 nm. The mechanism of fluorescence detection of organic phosphorus with CDs@VPA sensor might be attributable to hydrogen bonds formed between heteroatom O, N, S, or P, and the O−H group, which led to fluorescent quenching. Meanwhile, HN−C=O and Si−O groups in CDs@VPA system might contribute to cause excellent blue photoluminescence. The fluorescence sensor was thorough successfully employed to the detection of triazophos in cucumber samples, illustrating its tremendous value towards food sample analysis in complex matrix.

## 1. Introduction

Organophosphorus pesticides (OPPs) are the most extensively applied pesticides due to their high efficiency, broad spectrum, and low price, accounting for approximately 70% of the total pesticides [1]. Although low-toxicity OPPs are widely used, it is inevitable that exposed food is polluted by OPPs due to overuse, so that growing diseases caused by the excess accumulation of OPPs result in enormous injuries and deaths every year [2]. Thus, it is significant to establish a rapid analysis method with higher sensitivity and selectivity to detect OPPs for ensuring food safety and human health.

Nowadays, the detection strategies of OPPs mainly include spectroscopy, chromatography, immunoassays, enzyme inhibition, biochip technology, etc. [3]. Among them, the conditions for spectrophotometric determination are difficult to accurately grasp and the stability needs to be further improved [4]. Chromatography is one of the important methods to analyze OPPs for supervision departments. However, it requires large-scale instruments and fussy sample pretreatment, so it is not facile to rapidly detect OPPs on-site [5]. Immunoassays and biochip technology will be utilized in the near future, but their using in multi-residue detection will be limited due to their specificity [6,7]. In contrast, the fluorescent carbon-dot sensors, based on magnetic, molecularly-imprinted technology have the distinctive superiorities of fast response, strong specificity, and reusability, which make them suitable for rapid and real-time determination [8].

OPPs mainly have eight structural types: phosphonates, thiophosphonates, dithiophosphonates, phosphamides, thiophosphamides, pyrophosphoric esters, phosphonate esters, and thiophosphonate esters [9]. Although OPPs have diverse structures, we found that they all contain a phosphoric acid (or thiophosphorus acid) skeleton [10]. Carbon dots (CDs) with an approximate size of 20 nm are widely used in the field of fluorescence detection and analysis because of their good photoluminescence characteristics [11], low toxicity [12], green synthesis [13], biocompatibility [14], low cost [15], small size [16], and light bleaching resistance [17]. A large number of researchers have found that the carbon source for synthesizing carbon dots usually contains organic compounds with a conjugated system, such as C=N, C=O, or N=O groups, and strong, visible emission can be produced through cross-linking [18,19]. With profound research, it is gradually being realized that surface modification and heteroatom doping not only introduces reactive groups, but also improves the quantum yield [20]. Zhang et al. synthesized N-doped CDs with anhydrous citric acid and *N*-(beta-aminoethyl)-gamma-aminopropyl methyl dimethoxysilane by one-step hydrothermal method. The fluorescence efficiency of obtained CDs can be greatly improved by introduced N [21]. Therefore, we proposed that a magnetic, molecularly-imprinted polymer fluorescence-sensing system (magnetic, molecularly-imprinted polymeric microspheres-vinyl phosphate-modified carbon dots—MMIPs-CDs@VPA) could be constructed to recognize, enrich, and detect phosphoric acid skeleton to achieve a sensitive and rapid detection of OPPs.

In our study, we used Gaussian09W in a DFT/B3LYP/6-31G (*d, p*) group to screen the optimal functional monomers and their molar ratios to improve imprinting efficiency and specific adsorption of MMIPs for OPPs [22]. MMIPs are powerful materials to increase specific adsorption for target molecules in complex matrixes [23]. According to the screen of the molecular simulation, vinyl phosphate and 1-octadecene were identified as dual functional monomers, and then, surface imprinting polymerization on the surface of magnetic vinyl-modified Fe_3_O_4_@mSiO_2_ was applied to form a cavity in which geometric configuration and functional sites are identical with triazophos [24,25], so that obtained MMIPs could quickly and specifically recognize triazophos. N-doped carbon dots were prepared by one-step hydrothermal method [26], and modified by methacryloxy propyl trimethoxyl silane (MPS) to form double bonds. Subsequently, vinyl phosphate (VPA) was used as a surface modifier to prepare vinyl phosphate-modified carbon dots (CDs@VPA). After being enriched by MMIPs, triazophos in cucumber was eluted, and then detected by the CDs@VPA system, which could produce fluorescence quenching to realize a rapid, quantitative determination of triazophos. The constructed MMIPs-CDs@VPA (Figure 1) method for detecting OPPs has superiority in specific recognition and rapid detection over other detection analysis methods.

## 2. Materials and Methods

### 2.1. Chemicals and Reagents

Polyethylene glycol 6000 (PEG 6000), sodium acetate, FeCl_3_·6H_2_O, cetyltrimethyl ammonium bromide (CTAB), tetraethyl orthosilicate (TEOS), Anhydrous citric acid, 3-aminopropyl-triethoxy-silane (APTES), acetonitrile, ethanol, acetic acid, acetone, and HPLC grade acetonitrile were purchased from Sinopharm Chemical Reagent Co., Ltd. (Shanghai, China). Ethylene glycol dimethacrylate (EGDMA), α-methylacrylic acid (MAA), 4-vinylpyridine (4-VP), and methacryloxy propyl trimethoxyl silane (MPS) were acquired from Shanghai Macklin Biochemical Co., Ltd. (Shanghai, China). Vinyl phosphate (VPA), 1-octadecene (1-Oc), and quinalphos were purchased from Chem Baihua Mall (Shanghai, China). Triazophos, phosemet and *O*-ethyl-*O*-(4-nitrophenyl) phenyl thiophosphonate (EPN) were bought from Best-reagent (Chengdu, China). The 2,2′-azobis-isobutyronitrile (AIBN) was provided by Chinasun specialty products Co., Ltd. (Suzhou, Jiangsu, China). Triple distilled water, newly collected from a glass distillator (Shanghai Yarong Biochemical Equipment Co., Ltd., Shanghai, China), was used to prepare aqueous solutions.

### 2.2. Software for Molecular Simulation

Molecular simulation was operated through Gaussian09W and GaussianView (Version 5.0) using a workstation with Intel(R) Core(TM) i7-9750H (CPU @ 2.60 GHz), 16GB DDR4 (2666 MHz) and Window 10 (x64) (Gaussian Inc., Pittsburgh, PA, USA).

### 2.3. HPLC Analysis Conditions

HPLC analysis was conducted on the Beijing Purkinje L600 HPLC system equipped with a Pgrandsil-STC-C_18_ column (4.6 mm × 150 mm, 5 μm) (Beijing, China). A 20.0 μL sample was injected into the HPLC system with acetonitrile-water (55:45, *v*/*v*, for triazophos, phosemet and quinazophos, and 70:30, *v*/*v*, for EPN) as the mobile phase at flow rate 1.0 mL min^−1^, under the condition of the column temperature 30 °C, and detection wavelength of 247 nm (triazophos), 222 nm (phosemet), 237 nm (quinazophos), and 274 nm (EPN), respectively. The mixture of above four OPPs were detected at wavelength of 247 nm with acetonitrile (A) and water (B) as mobile phase in gradient elution: 0–16 min, 55% A; 16–16.5 min, 55%→60% A; 16.5–17 min, 60%→65% A; 17–17.5 min, 65%→70% A; 17.5–35 min, 70% A; 35–36 min, 70%→55% A; 36–46 min, 55% A.

### 2.4. The Synthesis of MMIPs, CDs, and CDs@VPA

The MMIPs were synthesized by the following procedure. The magnetic Fe_3_O_4_ particles, Fe_3_O_4_@mSiO_2_ particles, and vinyl-modified Fe_3_O_4_@mSiO_2_ particles were prepared in turn according to our previously published method [27,28,29], as follows. Firstly, FeCl_3_·6H_2_O (1.35 g), polyethylene glycol (1.00 g) and sodium acetate (3.60 g) were dissolved in 40 mL of ethylene glycol and ultrasonicated for 30 min to form yellow solution, which was then transferred into a 100 mL dry polytetrafluoroethylene lined autoclave and heated at 200 °C for 8 h. After cooling down, the mixture was washed 3 times by triple distilled water and ethanol, respectively, and then dried in vacuum at 50 °C for 8 h to obtain black Fe_3_O_4_ particles. Secondly, the above Fe_3_O_4_ particles (0.10 g) and CTAB (1.00 g) were added into 200 mL of triple distilled water and ultrasonicated for 30 min, which were slowly diluted with 900 mL of 1 mmol/L NaOH solution, ultrasonicated for 5 min, stirred at 60 °C for 30 min, then injected with 5 mL of TEOS/ethanol solution (1/4, *v*/*v*), continuously stirred for 5 min, and let stand for 12 h. Subsequently, the mixture was collected by magnet and then dispersed in 180 mL of acetone and refluxed at 50 °C for 48 h. After repeatedly washing 6 times with triple distilled water and drying under vacuum at 50 °C for 8 h, Fe_3_O_4_@mSiO_2_ particles were obtained. Thirdly, the Fe_3_O_4_@mSiO_2_ particles (0.25 g) were suspended in MPS solution (150 μL of MPS dissolved in 40 mL of 10% acetic acid solution) and stirred at 50 °C for 5 h, washed 3 times by triple distilled water, and dried under vacuum at 50 °C for 8 h to obtain vinyl-modified Fe_3_O_4_@mSiO_2_ particles.

Then, the selection and optimization of dual functional monomers for triazophos was conducted by simulation calculation with Gaussian09W, and then the screened dual functional monomers were validated through experimentation. In the molecularly-imprinted surface polymerization reaction, triazophos (0.25 mmol) and the four functional monomers MAA, 4-VP, VPA, and 1-Oc (0.25 mmol) were dissolved in methanol (12.0 mL), respectively, purged with N_2_, and then stored in a refrigerator at 4 °C for 12 h to prepare a preassembled solution. Vinyl-modified Fe_3_O_4_@mSiO_2_ (50.0 mg), EGDMA (5.0 mmol), and AIBN (20.0 mg) were dissolved in acetonitrile (15.0 mL) and the preassembled solution was added drop by drop, purged with N_2_ on ice, and then reacted for 24 h at 60 °C with constant stirring at 100 rpm [30]. Finally, the mesoporous silicon-based magnetic, molecularly-imprinted polymers (Fe_3_O_4_@mSiO_2_@MIPs, abbr. MMIPs) were washed by methanol-acetic acid (9/1, *v*/*v*) under stirring reflux at 70 °C to remove triazophos absolutely, which was monitored by HPLC. Then, the MMIPs were washed with methanol to pH = 7 before vacuum drying overnight at 50 °C. As a control, the same procedures were applied to prepare mesoporous silicon-based magnetic, molecularly non-imprinted polymers (Fe_3_O_4_@mSiO_2_@NIPs, abbr. MNIPs) in the absence of triazophos.

Not only the *Q*_MMIPs_ (see Section 2.5) of monofunctional monomers, but the *Q*_MMIPs_ of dual functional monomers were taken into account. Regarding *Q*, the molar amount of the template molecule triazophos and dual functional monomers were adjusted to obtain the optimal MMIPs.

As to CDs@VPA, the CDs were prepared according to the published procedure by Xu et al. [26]. Herein, anhydrous citric acid (0.5 g) was dispersed in APTES (5 mL), and the mixture was transferred into 25 mL of dry polytetrafluoroethylene lined autoclave purged with N_2_ for 2 min, and then reacted for 5 h at 210 °C. When cooling down, the color of the solution turned to yellow, and the crude product was centrifuged at 11,000 rpm for 1 min to obtain *N*-doped CDs, which were dispersed in a certain amount of ethanol and stored in the dark at 4 °C refrigerator for subsequent use. The procedure for preparing vinyl-modified CDs was similar to that for vinyl-modified Fe_3_O_4_@mSiO_2_.

After that, the CDs@VPA was synthesized using 0.1 mmol VPA dissolved in 6 mL ethanol in a 150 mL three-necked flask. Then, 16 mL of vinyl-modified CDs solution was added simultaneously, the mixture ultrasonicated for 2 min, purged with N_2_, and reacted at 60 °C for 24 h.

### 2.5. Adsorption Experiment of MMIPs

The adsorption capacity (*Q*) and imprinting factors (*α*) were calculated according to Equations (1) and (2), respectively.
*Q* = (*c*_0_ − *c*_e_) × *V*/*m*(1)
*α* = *Q*_MMIPs_/*Q*_MNIPs_(2)
where *c*_0_ and *c*_e_ represent the initial and the equilibrium concentrations (mmol L^−1^) of the solution, respectively. *V* is the volume (L) of the solution, and *m* represents the mass (g) of the MMIPs or MNIPs added. *Q*_MMIPs_ and *Q*_MNIPs_ are the adsorption capacity (mmol L^−1^) of the imprinted polymers and the corresponding non-imprinted polymers, respectively.

#### 2.5.1. Adsorption Kinetics and Thermodynamics of MMPs

Adsorption kinetics were carried out at 308 K under 150 rpm to investigate the variation of triazophos added to MMIPs and MNIPs over 240 min via the ratio (*m*/*v*) of 10 mg of the polymers to 10 mL of 1 mmol L^−1^ triazophos methanol solution. Adsorption thermodynamics were studied for 6 mg of the MMIPs added into 5 mL of the triazophos methanol solution by changing the initial concentration from 0.2–4 mmol L^−1^. Then, the series of polymers were shaken for 3 h at 298, 308, and 318 K, respectively. After separation by an external magnet, the concentrations of triazophos in the supernatant were detected by HPLC, and the adsorption capacity (*Q*) was calculated according to Equation (1).

#### 2.5.2. Adsorption Selectivity and Reusability of MMIPs

Selective adsorption experiments were implemented by adding 2 mL of mixed solutions (containing triazophos, phosemet, quinazophos, and EPN with the same initial concentration of 4 mmol L^−1^) to 10 mg MMIPs or MNIPs, and shaking for 3 h at 308 K under 150 rpm, respectively. The concentrations of the above-mentioned solution and the supernatant of adsorbed solution were detected by HPLC; then, the adsorption capacity (*Q*), imprinting factors (*α*), and separation factors (*β*) were calculated according to Equations (1)–(3), respectively.
*β* = *α*_triazophos_/*α*_others_(3)
where *α*_triazophos_ and *α*_others_ are the imprinting factors of triazophos and other structural analogues, respectively.

Reusability tests (recycle times) of MMIPs were carried out by adding 2 mL 4 mmol L^−1^ of triazophos solution to 10 mg MMIPs, and shaking for 3 h at 308 K under 150 rpm. The adsorption capacity of triazophos was calculated by Equation (1) after detecting the equilibrium concentrations by HPLC, as mentioned above. In each cycle, the adsorbed MMIPs were washed by methanol-acetic acid (9/1, *v*/*v*) to remove triazophos absolutely, which was monitored by HPLC. Then, the adsorption/desorption process repeated 7 times according to the above-mentioned procedures.

### 2.6. Structural Characterization of MMIPs, CDs, and CDs@VPA

The structures of the MMIPs were characterized by FT-IR, TEM, particle size analysis, and VSM. The structures of the CDs and CDs@VPA were characterized by FT-IR, UV-Vis, and TEM. Meanwhile, CDs were characterized by XPS. FT-IR spectra (4000–400 cm^−1^) were obtained via a Nicolet 6700 FT-IR spectrometer (Thermo Fisher Scientific, Boston, MA, USA). TEM (Tecnai G2 F20, FEI, Nashville, TN, USA) was used to observe diameter and morphology of particles, which compared with the results from particle size analysis (Nanoparticle size and Zeta potential analyzer, Malvern ZS90, London, UK). The magnetic property was measured at room temperature using VSM (Squid-VSM, Quantum Design, Atlanta, GA, USA). UV-Vis spectra was scanned and recorded on an UV2800 ultraviolet-visible spectrophotometry (Sunny Hengping Scientific Instrument Co., Ltd., Shanghai, China). XPS (Thermo Scientific Escalab 250Xi, Boston, MA, USA) was used to study the composition of substances.

### 2.7. Fluorescence Detection of Triazophos

Fluorescence (FL) detection (Fluorescence spectrophotometer, Agilent Technologies, model F4500, Los Angeles, CA, USA) of triazophos was carried out with the excitation wavelength 390 nm, excitation and emission slit widths of 5 nm, and a photomultiplier tube voltage of 600 V. The desiccated powder of CDs@VPA obtained (0.003 g) was dissolved into anhydrous ethanol (3 mL) to make an CDs@VPA mother solution. The 500 μL of triazophos standard solution or samples containing triazophos were dispersed into 100 μL of CDs@VPA mother solution above. After vortexing for 20 min, the fluorescence intensity of the mixture was measured.

For analysis of triazophos in fresh sample, the cucumber (5 g) was newly collected from farm and ground into suspension. After being filtered by 0.45 μm filter membrane, the filtrate was added triple distilled water to 25 mL. Then, 4 mL out of 25 mL filtrate was added into 20 mg MMIPs and shaken for 3 h. The adsorbed MMIPs were washed with triple distilled water and then eluted by methanol-acetic acid (9/1, *v*/*v*) and the eluent was detected for fluorescence intensity using above-mentioned method. According to the linear equation, the content of triazophos in cucumber was obtained. Meanwhile, the eluent was analyzed by HPLC to verify the accuracy of FL detection.

### 2.8. Data Analysis

All data were calculated by Origin software (version 8.0) for Microsoft Windows (OriginLab Corporation, Northampton, MA, USA).

## 3. Results and Discussion

### 3.1. Molecular Simulation

#### 3.1.1. Molecular Simulation to Screen Optimal Functional Monomer

Aiming at triazophos, the structures of 4-VP, MAA, VPA, and 1-Oc were optimized with Gaussian09W in DFT/B3LYP/6-31G (*d, p*) group, in which the binding energies (Δ*E*s) were calculated according to Equation (4). The ball-stick model with the lowest energy was obtained. The DFT method is more accurate, with a moderate amount of calculating to do, than the HF, self-consistent field method. The base group DFT/B3LYP/6-31G (*d, p*) was chosen due to its superior accuracy for calculating strong conjugate systems, its suitability to H→Cl atoms, and its polarization function for the accurate calculation of energy [31].
Δ*E* = |*E*_complex_ − *E*_template_ − ∑*E*_momomer_|(4)
where Δ*E*, *E*_complex_, *E*_template_, and ∑*E*_momomer_ are the binding energy (kJ mol^−1^), the lowest energy of formed complex after binding (kJ mol^−1^), the lowest energy of the template (kJ mol^−1^), and the sum of the lowest energy of all monomers (kJ mol^−1^), respectively.

To overcome the problem of the environmental pollution and toxicity of OPPs, it is necessary to develop a convenient and rapid, molecularly-imprinted, solid-phase extraction for the determination of OPPs. Thus, a data simulation was carried out to screen a functional-monomer-fitting template molecule before screening and optimizing functional monomers in experiments. Although Li et al. [22] had carried out relevant data simulation and confirmed the feasibility of the data simulation through experiments, we also conducted corresponding experiments to demonstrate the practicability of the simulation. In the preparation of MMIPs by molecularly-imprinted surface polymerization, template molecule and functional monomers were pre-assembled to form relatively stable hydrogen bonds [32]. According to the structure of triazophos, it can form hydrogen bonds with compounds containing carboxyl groups, hydroxyl groups, and pyridine rings, and π-π conjugation builds up to form heterocyclic complexes [33]. Hence, the choice of functional monomers is a key factor in the formation of specific binding sites, particularly in non-covalent MMIPs.

In the computer simulation method mentioned above, we screened four functional monomers for non-covalent imprinting to fit triazophos, and then the simulated optimized structures were assembled into Figure 2. The theory holds that two genres of functional monomers combined with a template could mutually promote the bond energy, so that the numerical value of binding energy of the complexes formed between a template and functional monomer could increase; thus, the structure of the polymers would be more stable, and the selectivity of the MMIPs would be improved, which was indeed confirmed by many studies [22,34,35,36]. As shown in Table 1, triazophos–VPA–1-Oc was the optimal complex with the highest binding energy. The functional monomer VPA could form hydrogen bonds and similar polarity attraction with OPPs molecules to enhance the order of intermolecular binding and form a structurally-ordered imprinted polymer [37]. The non-polar carbon chain of the weak polar functional monomer 1-Oc could interact with weak polar or non-polar functional groups in the OPPs molecules to form a hydrophobic interaction force, strengthen the specific adsorption of OPPs by molecular imprinting materials, and make the imprinting cavity more stable, and have stronger bonding and higher selectivity [38].

#### 3.1.2. Optimizing the Molar Ratio of Triazophos to VPA and 1-Oc

The poor predictability of the experiment and the lack of theoretical guidance during the preparation process are major problems in screening-imprinted systems. In recent years, with the developments of quantum chemistry and computer technology, molecular simulation technology has been applied to molecular imprinting to predict and confirm the binding of template-monomer complexes. The greater the binding energy, the more combined energy released, and the more easily they react, the more stable the complex that is formed [39]. On the basis of the optimal, functional monomers screened (VPA and 1-Oc) for triazophos, as mentioned above, the molar ratios of triazophos, VPA, and 1-Oc were also optimized by molecular simulation. The binding energy of the complex was highest (2174.7017 kJ mol^−1^) when the molar ratio of triazophos:VPA:1-Oc was 1:4:4 (Figure 3). With the molar ratio increasing, the Δ*E*s of complexes increased, whereas continual increasing of the amount of VPA and 1-Oc can cause the decreasing of the Δ*E* of complexes. Perhaps an excessive molar amount can cause the structure of the pre-assembled complexes which are too disordered to bond.

### 3.2. Optimization of the Synthesis of MMIPs

According to the adsorption effect of monofunctional, monomer-imprinted polymers (Table 2), both VPA–MMIPs and 1-Oc–MMIPs had higher adsorption capacities: 0.2435 mmol g^−1^ (*α* = 3.003) and 0.2599 mmol g^−1^ (*α* = 3.128), respectively. With VPA and 1-Oc as dual functional monomers, VPA+1-Oc–MMIPs were prepared. When the molar ratio of triazophos–VPA–1-Oc pre-assembled complex was 1:1:1 and 1:4:4, the adsorption capacity reached saturated state with the maximum *Q* = 0.2656 mmol g^−1^ (*α* = 3.3878) and *Q* = 0.3198 mmol g^−1^ (*α* = 4.059), respectively (Figure 4). Moderately increasing the amount of the dual functional monomers could raise the adsorption capacity of prepared MMIPs. Taking into account all the results of molecular simulation and experimental polymerization procedure, the optimal dual functional monomers for triazophos were VPA and 1-Oc. The optimum molar ratio of triazophos, VPA, and 1-Oc for preparing MMIPs was 1:4:4. That is to say, the data obtained from the molecular simulation above were reasonable.

### 3.3. Adsorption Study of MMIPs

Before studying the kinetics, thermodynamics, and selectivities of the MMIPs prepared, the concentrations for a standard series of four OPPs compounds were detected by HPLC, and then the standard curves were drawn. The linear regression equation, linear range, correlation coefficient (R^2^), limit of detection (LOD), limit of quantitation (LOQ), and relative standard deviation (RSD) were obtained, as shown in Appendix A.

#### 3.3.1. Kinetics of Adsorption on MMIPs

The adsorption kinetics of MMIPs and MNIPs at 308 K were studied to identify the adsorption rate and equilibrium time of target molecule triazophos on MMIPs. As shown in Appendix A, the adsorption capacity of MMIPs increased rapidly in the initial 130 min, and MNIPs increased rapidly in the initial 60 min. The adsorption of triazophos on MNIPs reached adsorption equilibrium at 90 min, while the adsorption of triazophos on MMIPs reached adsorption equilibrium at 180 min. Hence, 180 min was the optimal adsorption time. It took more time for MMIPs to achieve binding equilibrium than MNIPs, which could be attributed to the specific molecular recognition process on tailored stereo cavity and binding sites of MMIPs, and the physical adsorption of randomly-distributed functional groups on the surface of MNIPs [28].

A pseudo-first-order dynamic model and pseudo-second-order dynamic model were used to study the adsorption performance of the process, which were calculated according to Equations (5) and (6), respectively.
ln(*Q*_e_ − *Q*_t_) = ln*Q*_e_ − *k*_1_*t*(5)
*t*/*Q*_t_ = *t*/*Q*_e_ + 1/(*k*_2_*Q*_e_)(6)
where *Q*_e_ and *Q*_t_ represent the adsorption capacity (mmol g^−1^) at equilibrium and *t* moment; k_1_ and k_2_ are the rate constants of the two kinetics models, respectively.

According to the correlation coefficient (R^2^) shown in Appendix A, the adsorption of triazophos on the dual functional monomers MMIPs and MINPs conformed to a pseudo-second-order kinetics model because the R^2^ values of both were higher than 0.98. The calculated equilibrium adsorption value (0.160 mmol g^−1^) of MMIPs was close to the actual equilibrium adsorption value (0.167 mmol g^−1^) at 308 K. The results indicated that both external diffusion and intraparticle diffusion existed simultaneously [30].

#### 3.3.2. Thermodynamics of Adsorption on MMIPs

The adsorption of triazophos on MMIPs reached a maximum equilibrium adsorption capacity (0.226 mmol g^−1^) at 318 K through adsorption thermodynamics (Appendix A). With the increase of initial concentration, the adsorption capacities of all MMIPs increased under three diverse temperatures. The adsorption capacity increased with the increase of temperature; perhaps the reason was that the MMIPs could be swollen easily under higher temperature so that more binding sites were exposed. Thus, the adsorption capacity at a higher temperature was greater than that at low temperature, which is consistent with the results of relevant literature [28,30,40].

Langmuir and Freundlich adsorption isotherm models (Equations (7) and (8)) were applied to evaluate the affinity of binding sites of MMIPs at different temperatures. The parameters of the two models were expressed in Appendix A.
*c*_e_/*Q*_e_ = *c*_e_/*Q*_m_ + 1/(*Q*_m_*K*_L_)(7)
ln*Q*_e_ = *m*ln*c*_e_ + ln*K*_F_(8)
where *c*_e_ and *Q*_e_ are the equilibrium concentration (mmol L^−1^) and the equilibrium adsorption capacity (mmol g^−1^); *Q*_m_ is the saturated adsorption capacity (mmol g^−1^); and *K*_L_, *m* and *K*_F_ stand for the constants of the two models.

The Langmuir adsorption equation represented the monolayer adsorption process, while Freundlich’s adsorption equation introduced the intermolecular force, which represented the multilayer adsorption process. The correlation coefficient of the Langmuir adsorption model (0.987) in Appendix A was higher than that of Freundlich adsorption model (0.901), and its saturated adsorption capacity (0.238 mmol g^−1^) was closer to the experimental adsorption capacity (0.226 mmol g^−1^). Therefore, the thermodynamic adsorption behavior of MMIPs could be deemed to the monolayer adsorption.

#### 3.3.3. Competitive Selectivity and Reusability of MMIPs

For studying the adsorption selectivity of the MMIPs, triazophos was mixed with its three analogues (phosemet, quinalphos, and EPN) in methanol solution before the adsorption experiment. The adsorption capacity of triazophos on MMIPs was the maximum one, reaching 0.142 mmol g^−1^, while the adsorption capacities of phosemet, quinalphos, and EPN on MMIPs were only 0.034, 0.045, and 0.028 mmol g^−1^, respectively (Appendix A). Whereas there were no distinct differences in the adsorption capacities of MNIPs for the four analogues, and the imprinting factor of MMIPs *α*_MMIPs/MNIPs_ = 4.59 for triazophos, for the separation factors: *β*_triazophos/phosemet_ = 3.73, *β*_triazophos/quinalphos_ = 3.23, and *β*_triazophos/EPN_ = 3.79; they are reflections of the adsorption separation abilities of the MMIPs between triazophos and the analogues; thus, the MMIPs show better selectivity for triazophos than their three analogues.

Compared with the MAA+4VP–MMIPs synthesized by Li et al. [22], kinetic adsorption for the MMIPs to zearalenone reached adsorption equilibrium with the value of 26.875 μg mg^−1^ (equal to 0.085 mmol g^−1^), whereas our proposed VPA+1-Oc−MMIPs to triazophos had an adsorption equilibrium value of 0.167 mmol g^−1^ under the same conditions. Considering competitive selectivity, the imprinting factor of MAA+4VP–MMIPs *α*_MMIPs/MNIPs_ = 2.10, and the biggest separation factor was β_zearalenone/amlodipine_ = 1.85; however, the proposed VPA+1-Oc−MMIPs to triazophos had α_MMIPs/MNIPs_ and β_triazophos/EPN_ 2.18 and 2.05 times those of MAA+4VP–MMIPs to zearalenone, respectively. Therefore, our proposed MMIPs have certain advantages in terms of adsorption efficiency and selectivity.

Appendix A explained the reusability of MMIPs. MMIPs showed higher reusability because the adsorption capacity of the MMIPs was maintained at 94.5% after recycling seven times. All the above-mentioned results proved that the MMIPs synthesized by our proposed method possessed high specific adsorption, selectivity, and reusability.

### 3.4. Characterization of MMIPs

The results from TEM and particle size analysis indicated that the Fe_3_O_4_ particles, mesoporous SiO_2_-modified Fe_3_O_4_ particles (namely, Fe_3_O_4_@mSiO_2_), and MMIPs were quite polydisperse structures with an almost spherical morphology (Figure 5). Meanwhile, Fe_3_O_4_ particles had an average diameter of 335 nm. Fe_3_O_4_@mSiO_2_ particles had an average diameter of 375 nm, probably due to being enveloped by the silica layer. MMIPs had a typical core-shell structure with an average diameter of 430 nm. It can be concluded that core-shell molecularly imprinted polymers have been successfully prepared.

In FT-IR (Figure 6), the appearance of the characteristic peak at 582 cm^−1^ represented the vibration of Fe–O (Figure 6a–f). It can be seen that 582 cm^−1^ absorption peak represented Fe_3_O_4_ (Figure 6a). There was a strong peak at 1059 cm^−1^ (Figure 6b–f), which was the Si–O stretching vibration peak, and there were characteristic absorption peaks of CTAB at 2917 and 2845 cm^−1^ for Fe_3_O_4_@CTAB/mSiO_2_ (Figure 6b), whereas there were no characteristic absorption peaks of CTAB for Fe_3_O_4_@mSiO_2_ (Figure 6c). These results demonstrated that CTAB was absolutely eluted, and combining Figure 5a,b, the mesoporous silica was formed and successfully coated on the surface of Fe_3_O_4_. As for Figure 6d, the peak at 1633 cm^−1^ represented a C=C double bond and the absorption peak of the C=O double bond at 1718 cm^−1^ appeared in Figure 6d–f, which indicates that the conjugated double bonds rooted in MPS were successfully grafted onto Fe_3_O_4_@mSiO_2_ surface (namely, vinyl-Fe_3_O_4_@mSiO_2_), which would enhance the chemical and mechanic stability of MMIPs (Figure 6e) or MNIPs (Figure 6f), and reduce the difficulty of polymerization. The intensity of Fe–O and Si–O stretching vibration peaks in Figure 6d compared to Figure 6e–f were slightly decreased, possibly due to the existence of the polymer layer. All the results of FT-IR, TEM, and particle size analysis of Fe_3_O_4_@mSiO_2_ and MMIPs demonstrated that the polymer layer had successfully adhered to the surface of Fe_3_O_4_@mSiO_2_.

In the vibrating sample magnetic strength measurement (VSM, Appendix A), the shape of magnetization curves of Fe_3_O_4_, Fe_3_O_4_@mSiO_2_, and MMIPs was similar; there was no remanence and the coercivity value was zero, which indicated that three materials were superparamagnetic. At room temperature, the saturation magnetizations of Fe_3_O_4_, Fe_3_O_4_@mSiO_2_, and MMIPs were 75.4 emu g^−1^ (Appendix A), 67.5 emu g^−1^ (Appendix A), and 65.9 emu g^−1^ (Appendix A), respectively, and the magnetic intensity decreased in turn. The reason was that non-magnetic SiO_2_ and polymer layers were coated on the surface of Fe_3_O_4_ and Fe_3_O_4_@mSiO_2_, which would weaken the magnetism. Besides, we found that the saturation magnetization of Fe_3_O_4_ prepared by solvothermal reaction was stronger than that of synthesized by coprecipitation. The magnetron effect (Appendix A) proved that the MMIPs obtained had high superparamagnetism and could be easily separated by external magnets.

### 3.5. Optimization of CDs and CDs@VPA Synthesis

Molar ratio, reaction temperature, and reaction time values of anhydrous citric acid and APTES can affect the fluorescence behavior of CDs. Hence, preparation conditions should be optimized to achieve better fluorescence properties. Here, the volume of APTES was set as 5.00 mL, and the mass of anhydrous citric acid was changed to investigate the effect of different molar ratios of anhydrous citric acid on APTES (1:6.8–1:11.7) in terms of fluorescence intensity and quenching efficiency (Appendix A). The maximum fluorescence intensity was obtained, and the quenching efficiency significantly increased when the molar ratios of anhydrous citric acid to APTES increasing from 1:6.8 to 1:8.2, and then fell off afterwards. Moreover, reaction temperature (130–250 °C) and reaction time (2–7 h) were also considered (Appendix A). Too high temperature and long reaction time would damage the surface active group [41]. Obviously, CDs by solvothermal method at 210 °C for 5 h have relatively higher fluorescence intensities and quenching efficiencies.

Similarly, the optimization of the synthesis conditions for CDs@VPA was carried out as for CDs. When the molar ratio was 1:4 vinyl-CDs to VPA and the reaction was at 60 °C for 24 h (Appendix A), the fluorescence intensity of CDs@VPA was strongest. The pH and incubation time of detection also play important roles on chemical stability and photostability of CDs@VPA in the fluorescence detection. The influence of pH on the fluorescence intensity of CDs@VPA was investigated (Appendix A), for which fluorescence intensity increased with the pH from 4 to 7 but decreased afterwards, indicating that the optimal pH of CDs@VPA was 7. Setting pH at 7, the stability of CDs@VPA solution was also investigated. It was found that the quenching equilibrium of CDs@VPA solution was achieved in 10 min, and it still remained stable although incubated up to 50 min (Appendix A). Thus, the optimal detection pH was 7 and incubation time of CDs@VPA was 10 min.

### 3.6. Characterization and FL Test of CDs@VPA

#### 3.6.1. Characterization of CDs and CDs@VPA

The morphology of CDs and CDs@VPA was characterized by TEM. The CDs (Figure 7a) and CDs@VPA (Figure 7b) both displayed spherical structures with mean diameters of about 50 and 120 nm, respectively. XPS spectra of CDs (Figure 7c) indicated that CDs contain carbon, nitrogen, and oxygen atoms with contents of 65.44%, 9.44%, and 25.11%, respectively. The C 1s spectra (Appendix A) indicated the existence of C=C/C–C (284.36 eV), C–N (285.01 eV), C–Si (285.91 eV), and C=O/C–O (287.98 eV); N 1s spectra (Appendix A) confirmed the presence of N–H (399.22 eV) and C–N (399.89 eV); and O 1s spectra (Appendix A) displayed four functional groups, including O–H (531.26 eV), C=O/C–O (531.90 eV), and O–Si (532.59 eV) [42,43,44]. The XPS spectra demonstrated that APTES was successfully bonded to anhydrous citric acid.

FT-IR was used to validate the bonding condition of surface chemical functional groups at different synthesis stages (Figure 8a). All of CDs, vinyl-CDs, and CDs@VPA had some similar peaks at 1588 and 1080 cm^−1^, which were attributed to the bending vibrations and stretching vibrations of N–H and Si–O, respectively. Additionally, the absorbance peaks at 1716 cm^−1^ in CDs represented the stretching vibrations of C=O of a straight-chain secondary amide, indicating the formation of amide groups was proven by the analysis results from XPS (Figure 7c and Appendix A). Another absorption peak at 1617 cm^−1^ represented the stretching vibrations of C=C, which verified that the CDs was successfully modified by MPS. The absorbance peaks at 2409 cm^−1^ represented the stretching vibrations of P–OH, which was the characteristic absorption peak of VPA. The broad peaks in the range of 2500–3686, 2866–3006, and 3309–3499 cm^−1^ were related to the stretching vibrations of O–H, C–H, and N–H, respectively, demonstrating the VPA was bonded on the surface of CDs [45].

The optical properties of CDs and CDs@VPA were also studied. CDs were assessed by UV–Vis absorption spectrum and fluorescence spectra (Figure 8b). There was an absorption peak situated at 364 nm in UV–Vis absorption spectrum. When the excitation wavelength ranged from 350 to 460 nm, the emission wavelength showed red shifts, along with the reduction of photoluminescence energy in the fluorescence emission spectra [46]. The fluorescence intensity gradually enhanced with the excitation wavelength increasing from 350 to 390 nm, and there was the highest fluorescence intensity at 390 nm. Afterwards, the fluorescence intensity decreased with increasing of excitation wavelength from 410 to 460 nm. It was revealed that the emission spectra intensively relied on excitation wavelength. The fluorescence excitation and emission spectra (Figure 8c) of CDs@VPA showed a strong blue fluorescence with the ultraviolet excitation illumination due to enormous O–H, HN–C=O, and Si–O groups on its surface, and presented a symmetrical shape of excitation and emission spectra [47]. The fluorescence intensity of CDs@VPA weakened slightly in 8 days when stored in the dark at 4 °C in a refrigerator for subsequent gauging (Figure 8d). Perhaps the silane functional group from APTES coated on the surface of CDs caused the fluorescence intensity to be much more steady [48].

#### 3.6.2. Measurement of Triazophos by CDs@VPA in Cucumber

The reaction system of CDs@VPA contained HN−C=O and Si−O groups, making it possess excellent blue photoluminescence. Thus, the CDs@VPA system could detect organic phosphorus rapidly by fluorescence quenching with fluorescence spectrophotometry. That quenching mechanism, rooted in the O–H of VPA, could involve forming hydrogen bonds with heteroatoms N, O, S, and P of OPPs [49]. Figure 9a shows a trend, that the fluorescence intensity of CDs@VPA decreased with the concentration of triazophos increasing. The quenching levels of fluorescence intensity could be quantitatively described by Stern–Volmer equation as follows:*F*_0_/*F* = 1 + K_SV_ × *c*(9)
where *F*_0_ and *F* are the fluorescence intensity in the absence and presence of quencher, respectively; K_SV_ is the quenching constant; and *c* is the concentration of triazophos (mmol L^−1^).

The fluorescence quenching property of CDs@VPA by triazophos with different concentrations from 0 to 0.20 mmol L^−1^ are shown in Figure 9a. Figure 9b presents a linear relationship between the value of *F*_0_/*F* on CDs@VPA and concentration of triazophos in the range of 0 to 0.20 mmol L^−1^ (R^2^ = 0.9988). The LOD was 0.0015 mmol L^−1^, which was calculated according to the 3 σ/K criterion (σ is the relative standard deviation 0.00347 of blank measurements (n = 10), and K is the slope 6.9371 of linear fitting graph (Figure 9b), which is similar to that from other detection techniques for different analytes [26].

Compared to CDs@VPA fluorescence detection method pretreated by MMIPs, at the same time, the triazophos in cucumber sample were analyzed by HPLC, and the results of the two methods are close to each other (Table 3). Thus, the CDs@VPA fluorescence detection with pretreatment by MMIPs could be used to analyze the triazophos in food.

## 4. Conclusions

In conclusion, this study offered a comprehensive molecular simulation strategy by calculating the lowest energy conformation of complexes between template molecule and functional monomers to screen optimal functional monomers before polymerization. VPA and 1-Oc could be regarded as the optimal dual functional monomers to produce molecularly-imprinted materials for recognizing OPPs. Using surface-imprinting polymerization, the dual-functional monomers’ magnetic, molecularly-imprinted polymers (VPA+1-Oc−MMIPs) and four, single-monomer MMIPs were synthesized to verify the practicability of the molecular simulation of Gaussian09W in DFT/B3LYP/6-31G (*d, p*) group. After enriching OPPs by MMIPs, OPPs could be rapidly detected in 2 min with fluorescence quenching by a novel fluorescent material CDs@VPA with highly fluorescence property. Due to its phosphoric acid-like skeleton, the CDs@VPA system could be utilized to detect almost all OPPs, once using other OPPs as templates. As enrichment elements, MMIPs possessed highly specific adsorption, selectivity, reusability, and could be utilized to concentrate extremely dilute OPPs solution. With the superiority of having a simple, low-consumption synthesis and rapid detection, the CDs@VPA could be a prospective fluorescence sensor to detect organic phosphorus in real samples.

## Figures and Tables

**Figure 1 polymers-11-01770-f001:**
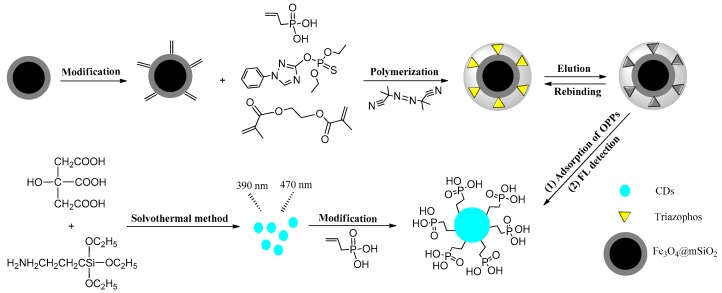
Illustration of the preparation of magnetic, molecularly-imprinted polymeric microspheres-vinyl phosphate-modified carbon dots (MMIPs-CDs@VPA) detection system for triazophos.

**Figure 2 polymers-11-01770-f002:**
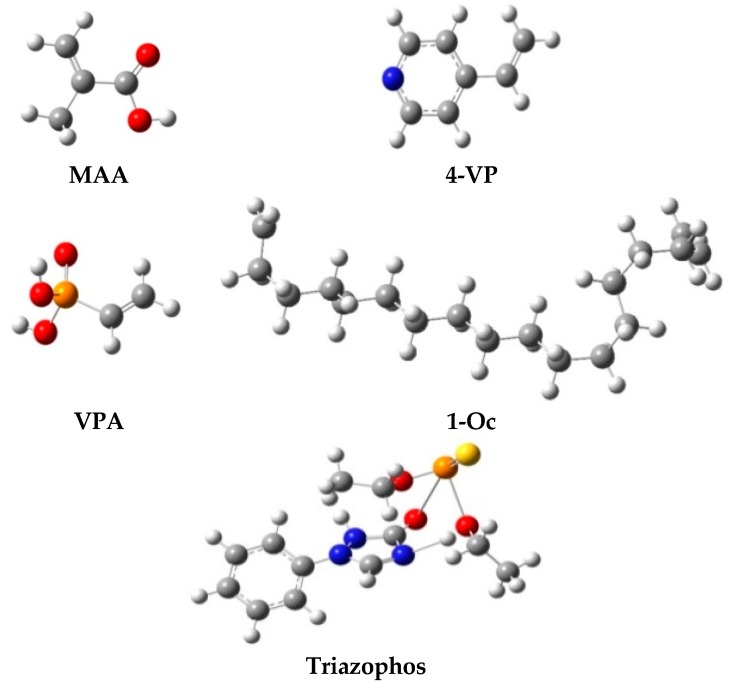
Ball and stick model of four candidate functional monomers and triazophos for structural optimization.

**Figure 3 polymers-11-01770-f003:**
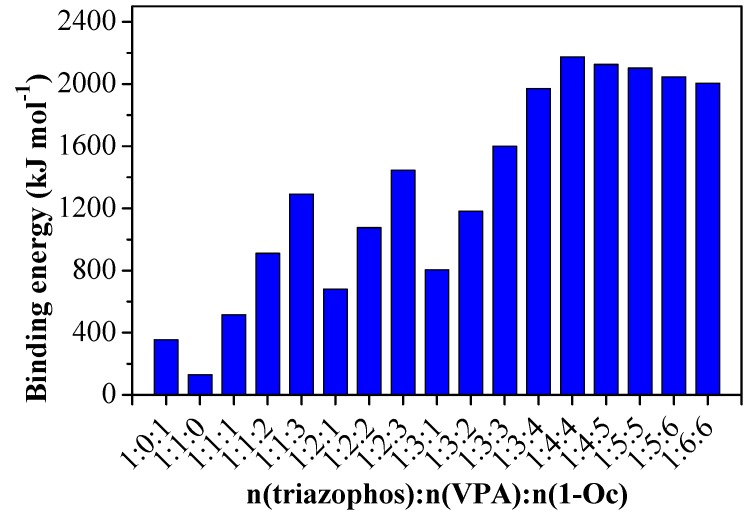
Binding energy (Δ*E*) between triazophos to VPA and 1-Oc under different molar ratios calculated by Gaussian molecular simulation.

**Figure 4 polymers-11-01770-f004:**
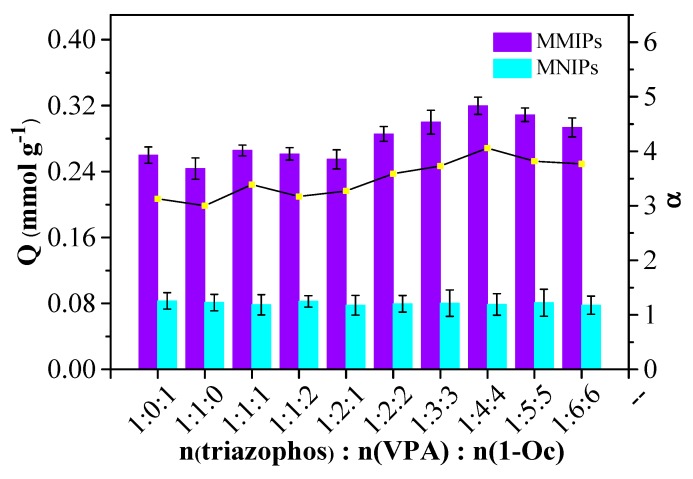
The effect of the mole ratio of template and dual-functional monomers′ pre-assembled complexes on adsorption capacities of MMIPs and MNIPs.

**Figure 5 polymers-11-01770-f005:**
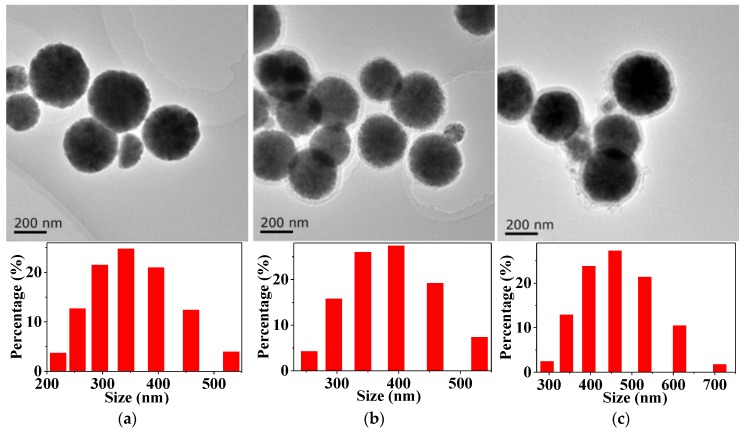
TEM and particle size analysis of Fe_3_O_4_ (**a**), Fe_3_O_4_@mSiO_2_ (**b**), and MMIPs (**c**).

**Figure 6 polymers-11-01770-f006:**
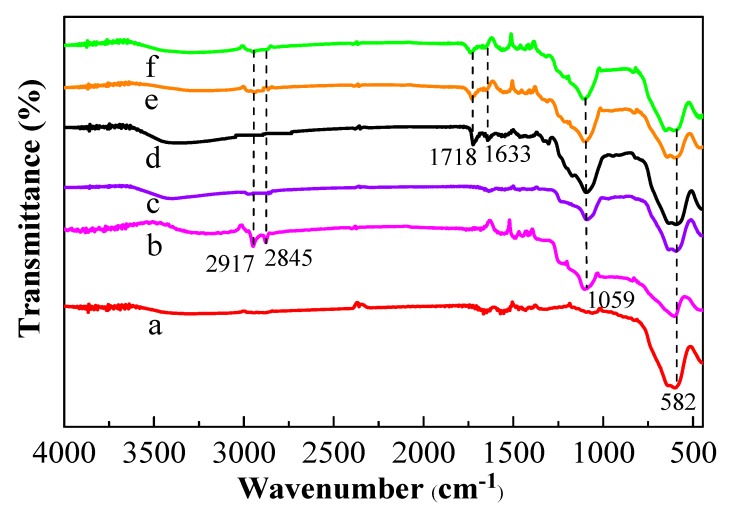
FT-IR spectra of Fe_3_O_4_ (**a**), Fe_3_O_4_@CTAB/mSiO_2_ (**b**), Fe_3_O_4_@mSiO_2_ (**c**), vinyl-Fe_3_O_4_@mSiO_2_ (**d**), MMIPs (**e**), and MNIPs (**f**).

**Figure 7 polymers-11-01770-f007:**
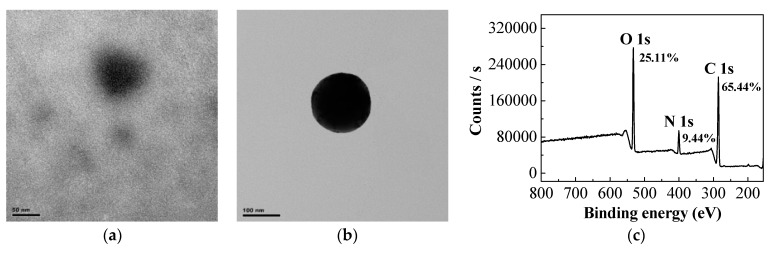
TEM of CDs (**a**) and CDs@VPA (**b**); XPS spectra of CDs (**c**).

**Figure 8 polymers-11-01770-f008:**
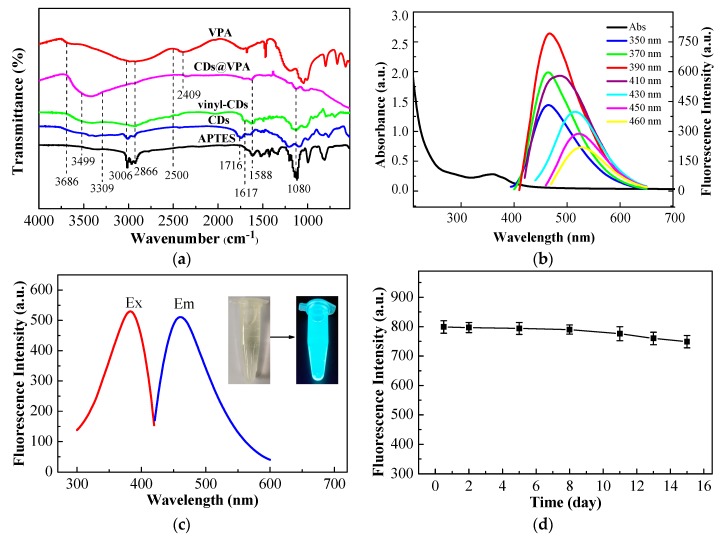
FT-IR spectra of APTES, CDs, vinyl-CDs, CDs@VPA, VPA, and CDs@VPA (**a**); absorption and fluorescence spectra of CDs (excitation wavelength from 350 to 460 nm; 8 mg mL^−1^ CDs in all analyte solution) (**b**); fluorescence excitation and emission spectra of CDs@VPA (**c**); the fluorescence stability property of CDs@VPA from 0.5 to 15 days at 4 °C refrigerator in the dark (**d**).

**Figure 9 polymers-11-01770-f009:**
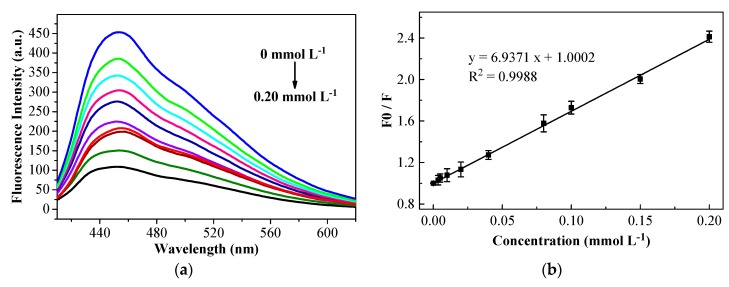
Fluorescence spectra of CDs@VPA in the presence of various concentrations of triazophos: 0, 0.0035, 0.005, 0.01, 0.02, 0.04, 0.08, 0.10, 0.15, and 0.20 mmol L^−1^, respectively (**a**). Stern–Volmer plot of triazophos concentration and the fluorescence intensity of triazophos-CDs@VPA complex (the error bars represent the standard deviations of three parallel tests) (**b**).

**Table 1 polymers-11-01770-t001:** The minimum energy of four candidate functional monomers, and triazophos and the binding energies of the complexes.

Compounds	Minimum Energy (Ha)	Binding Energy (Ha)	Binding Energy (kJ mol^−1^)
MAA	−306.4925	/	/
4-VP	−325.6957	/	/
VPA	−646.3007	/	/
1-Oc	−706.9781	/	/
Triazophos	−1597.6121	/	/
Triazophos - MAA	−1904.1237	0.0191	50.1471
Triazophos - 4-VP	−1923.3311	0.0232	60.9116
Triazophos - VPA	−2243.9627	0.0499	131.0125
Triazophos - 1-Oc	−2304.7253	0.1351	354.7051
Triazophos-MAA-4-VP	−2229.8741	0.0738	193.7619
Triazophos-MAA-VPA	−2550.5043	0.0990	259.9245
Triazophos-MAA-1-Oc	−2611.2362	0.1535	403.0143
Triazophos-4-VP-VPA	−2569.7154	0.1069	280.6660
Triazophos-4-VP-1-Oc	−2630.4462	0.1603	420.8677
Triazophos–VPA–1-Oc	−2951.0872	0.1963	515.3857

**Table 2 polymers-11-01770-t002:** Experimental verification of the adsorption capacity of imprinted polymer formed by triazophos and one out of four candidate functional monomers.

Monomer	*Q* (mmol g^−1^)		*α*
	MMIPs	MNIPs	
MAA	0.2209	0.0898	2.460
4-VP	0.2315	0.0902	2.567
VPA	0.2435	0.0811	3.003
1-Oc	0.2599	0.0831	3.128

**Table 3 polymers-11-01770-t003:** HPLC and MMIPs-CDs@VPA FL analysis of triazophos in cucumber sample.

Detection Method	Linear Range (mmol L^−1^)	LOD (mmol L^−1^)	Detection Time (min)	Concentration of Triazophos (mmol kg^−1^)
HPLC	0.0006–5.0	2.0 × 10^−6^	30	0.0049 ± 0.0011 *
MMIPs−CDs@VPA FL	0.0035–0.2	0.0015	2	0.0044 ± 0.0018

* The measured concentration error of triazophos refers to the standard deviations of three parallel tests.

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
