# Peer review of "Vinyl Phosphate-Functionalized, Magnetic, Molecularly-Imprinted Polymeric Microspheres’ Enrichment and Carbon Dots’ Fluorescence-Detection of Organophosphorus Pesticide Residues"

_polymers, 2019, doi:10.3390/polym11111770_

Round 1

Reviewer 1 Report

The manuscript remains of very poor quality. Experiments are described to the bare minimum which prevents another researcher from reproducing them. The synthesis of the particles must be better described, showing one example in details.

I suspect that the authors expect the reader to read all their papers so that the reader becomes familiar with their work. This is not the idea. The authors must provide sufficient information in their manuscript for the read to build an idea about the work without having to review tens of previously published by the authors. A manuscript must be a self-contained document, which I find, is not the case.

Most importantly, the detection limit obtained from the fluorescence data is absolutely not convincing. None of the data points actually lands on the Stern Volmer line. The intercept is off by 4% (1.04 instead of 1.00 for zero quencher) which implies that the authors have no accuracy in the low concentration range that they wish to study. The authors need to address this point seriously, by focusing on the fluorescence response of their sensors in the low concentration that they claim to study.

Author Response

Detailed comments can be found in the annex.

Reviewer 2 Report

The authors report the design, synthesis, and characterization of a molecularly imprinted polymer sensors for the fluorescent detection of the pesticide triazophos. 

The material was designed with a computation study. 

In addition to the sensing properties the synthesized materials were characterized with FTIR, XPS, and TEM.  The materials were shown to have a detection limit of 0.001 mM.  Tests with food samples gave similar results (accuracy and precision) to a referee HPLC method.  Furthermore the materials were shown to exhibit minimal degradation when stored or reused.

Some issues that should be addressed include:

The article references two articles on the use of molecularly imprinted polymers for the analysis of organophosphorus pesticides, but no meaningful analysis of these articles is given.  Why are there no comparisons on selectivity, absorption, kinetics, etc?  These could be helpful to evaluate the material. Reference 10 does not relate to the present article.  It is from a medical journal (frontiers in endocrinology) and does not discuss the structures of organophosphorus pesticides.

Some minor issues that should be addressed include the following:

Figure 1 - There appears to be a typo in the figure legend for triazophos Figure 8b and 9b - The figure captions should include details on what the error bars represent and how many measurements were used in the analysis. Table 3 - The caption should mention what the error on the measurement indicates. Figure 5 - The plots are very small, and difficult to read. Reference 1 - There is a [J] at the end of the reference that appears to be unnecessary.

Author Response

Detailed comments please see the attachment.

Reviewer 3 Report

What is the novety of presented work in comparison to the literature? What are advantages and disadvantages of proposed materials? In Figure 5 TEM images are absent

Author Response

(The authors gave the same response as above.)

Round 2

Reviewer 1 Report

See attached file.

Author Response

This manuscript is a resubmission of an earlier submission. The following is a list of the peer review reports and author responses from that submission.

Round 1

Reviewer 1 Report

see attached file

Reviewer 2 Report

The paper describes a method of collecting mmol/g amounts of organophosphate pesticide on a MMIP prior to desorption from the MMIP for detection and quantification by fluorescence spectroscopy via carbon dots. The work is very interesting and the method appears sound. I have the following comments and questions which may help the authors:

1) Lines 72-86 are quite complex and much of the information is repeated in the experimental section. I would recommend simplifying the description of the collection and measurement methods in clear short sentences - the procedure is not immediately clear.

2) Figure 1 should be bigger.

3) Lines 322-325 - could the adsorption capacity be increasing with temperature due to swelling exposing more binding sites?

4) Line 375 appears to have a word missing between "would" and "with".

5) Lines 379-380 - were the CDs stored in the dark/refrigerator etc between measurements?

6) Line 399 - it should perhaps be pointed out that this LoD being similar to other detection techniques are for different analytes...

7) Figures 7d and 8b seem to have been switched.

8) In the introduction on lines 47-48 large-scale instruments and fussy preparation are mentioned as challenges in real-world pesticide sensing, which is true. However, the method in this manuscript also requires access to expensive bench-top instruments, and the sample preparation and measurement are quite complex. How do the authors anticipate this method becoming practical for real-world use?

Round 2

Reviewer 1 Report

The manuscript Wu et al. on magnetic molecularly imprinted polymeric microspheres should not be published. Despite my suggested changes, the English of the manuscript remains horrendous. The simulation work, while it appears to be an accepted practice in the literature, is ridiculous, since vinyl monomers with carbon-carbon double bonds are employed when the double bond does not exist in the polymer matrix. While the authors have reduced the number of significant digits in their results, they still refuse to provide error bars for the vast majority of their results. The particles that they generate are fairly polydisperse which implies that they cannot use the size distribution to support a tiny increase in size imparted by coating the surface of the particles with a silica layer. Yet they continue doing so. Something remains amiss with the fluorescence quenching experiments shown in Figure 8b where the authors obtained a perfect straight line that does not pass through unity, which it should. At low triazophos concentration, Io/I is 4% off, but the authors insist that a minuscule limit of detection can be achieved when their experimental results are 4% off.